# A Simple Model to Compute the Characteristic Parameters of a Slotted Rectangular Microstrip Patch Antenna

**Oscar Ossa-Molina** [1,2,*,†,‡] **and Francisco López-Giraldo** [2,*,‡]

1    Electrical Department, Institución Universitaria Pascual Bravo, Medellín 050001, Colombia
2    Department of Electronics and Telecommunications Engineering, Instituto Tecnológico Metropolitano, Medellín 050013, Colombia
*    Correspondence: oscar.ossa@pascualbravo.edu.co (O.O.-M.); franciscolopez@itm.edu.co (F.L.-G.); Tel.: +57-3194648768 (O.O.-M.)
†    Current address: Cl. 73 # 73a-226, Medellín 050034, Colombia.
‡    These authors contributed equally to this work.

**Abstract:** In this study, we developed an analytical model of slot-loaded rectangular microstrip patch antennas based on the simulation results by varying slot parameters. The dominant resonant frequency predicted by such a model is in strong agreement with the experimental results measured at several locations of slot-loaded rectangular microstrip patch antennas. The model enables a reliable and quick computation of the resonant frequency, which generally follows a harmonic behavior that nearly resembles the resonant frequency of a microstrip antenna without a slot, which can be related to a small change in the impedance caused by the slot position. Results showed a good agreement between simulations and measurements for all the slot positions. Mathematical analytic functions to describe the changes in specific characteristic parameters of the slot-loaded rectangular microstrip patch antennas are also included.

**Keywords:** microstrip antennas; slot antennas; reflection coefficient; antenna analysis; antenna measurement; antenna theory; computer aided analysis

## 1. Introduction

The slot-type of discontinuities on diverse kinds of patch antennas has been investigated for more than four decades due to their usefulness in various fields of application as well as the savings they would produce in terms of cost, size, and weight [1]. There exists in the literature different seminal studies regarding slots on dielectric materials, which have been proposed with theoretical and experimental approaches primarily to improve antenna performance by adding slots [2–4]. For example, modification of the resonant mode of a rectangular patch antenna [5], achieving dual-polarization in a microstrip antenna [6], dual-frequency patch antennas [7], circularly polarized small-size microstrip antenna [8], broadband frequency response [9], etc. More recently, other applications of slots on the designing stage have been substantially proposed for solving several issues in communication, sensing, and antenna performance. For example, increased operating bandwidth antennas have been achieved by introducing slots within the antenna structure [10–12]. A narrowband configurable rotator was designed by using a frequency selective surface (FS) [13]. This FSS device was based on a circular substrate integrated waveguide (SIW) cavity with a slot on the front and two orthogonal slots on the back. Another work proposed a new design of a slotted circular patch antenna loaded with a metasurface that improved the system performance by allowing a total exposure of the solar cells to sunlight [14]. This antenna was fabricated with a lattice of $4 \times 4$ square solar cells. In the same year, a microfabricated point symmetric meander-line (ML) with slots inserted on the ground plane was presented to reduce the mutual coupling between narrowly spaced patches in a $2 \times 1$ antenna array [15]; this antenna array was intended to work in WLAN applications.

Other works have involved slots in biomass permittivity determination [16], a dual-band microstrip antenna with enhanced gain for energy harvesting [17], miniaturized UHF antennas [18], and a dual-band antenna for wearable applications [19].

Previous works have been typically analyzed using a full-wave numerical solver like commercial CAD/CAE software such as COMSOL, Computer Simulation Technology (CST), or ANSYS. Additionally, optimization algorithms, as well as computer-aided tools, are employed to optimize the designs of the antennas and meet desired goals. Several works have been proposed to overcome some limitations of previous modeling approaches such as complexity or accuracy. Authors have developed a modeling method based on the volume integral equation-based modeling method suitable for a patch or slot antenna on a thin finite dielectric substrate. They noted that results showed a small error by comparison with fine finite-element method (FEM) simulations, measurements, or the analytical mode-matching approach [20]. A new approach for antenna model optimization to obtain the infinitesimal dipole model (IDM) was proposed by applying the space map (SM) concept [21]. In later work, authors identified a way to overcome a computational complexity issue caused by the combination of IDM and full-wave analysis on the optimization process of finite ground plane antenna. So, using IDM and SM optimization they achieved lower simulation runtime compared to an optimization using a complete antenna model. An extended model of coax-fed printed metasurface antenna, which allows computing the input impedance of the metasurface and other key antenna parameters such as gain and efficiency has been presented [22]. A study on the impact of slot parameters on the first three resonant frequencies of a patch antenna has been proposed to find a relation between the slot's parameters and antenna performance [23]. In their results, the authors showed that the slot has a different influence on each resonant frequency mode, and the frequency changes showed unequal sensitivity to the variation of the given slot parameters. Moreover, changes in resonant frequencies would exhibit unintuitive results. Slots also have been used in flexible antennas to achieve monopole-like radiation patterns and pattern reconfigurability. For example, an ultrawideband antenna with monopole-like radiation patterns was achieved by using two arranged rings concentrically around the main annular-ring circular patch antenna [24]. A similar approach presented a conformal antenna with electronically tuning capability of its radiation. The reconfigurability is achieved by activating and deactivating the slots using PIN diodes, to switch between TM02 (monopole-like mode) and perturbed TM02 distributions (broadside mode) of the antenna [25]. Additionally, pair rectangular slots were recently proposed for extending the axial ratio in patch antennas as explained in [26].

In this paper, we extend [23] to the analysis of the influence of the rectangular slot on the resonant frequency belonging to the dominant (TM010) mode of a patch antenna. Our results answered those questions incorporated in [23] regarding the number of resonant frequency changes with the length, width, and slot position; and how sensitive the resonant frequency is to a particular slot parameter. Moreover, we found an expression that can describe the impact of slot parameters on the value of the resonant frequency under consideration. This paper is organized as follows. In Section 2, we describe the antenna design, the parametric analysis of the slot geometry modifications, the fabrication process, and the experimental setup. In Section 1, we start describing the antenna design, the parametric analysis of the slot, and the experimental setup. In Section 2, we present results from simulations and measurements. Thus, a comparison between them is also presented. Additionally, we introduce a model that describes the changes in antennas' parameters due to slot geometry. Finally, in Sections 4 and 5, we conclude from our findings and propose future works.

## 2. Antenna Design and Experimental Setup

In this section, we describe the modifications applied to a rectangular patch antenna in order to establish a relationship between the antenna's geometry and its electrical response. Modifications concern variations in both position and width of single rectangular

slot added to the radiating patch. Several parametric simulations were performed given specific slot positions and widths. Next, we analyze antenna parameter changes due to the slot, such as the resonant frequency, the magnitude of the reflection coefficient, the input impedance, the surface current, and the electric field. Finally, we fabricated six antennas to validate the simulation results.

### 2.1. Antenna Design and Configuration

A simple and usual method of analysis for conventional microstrip antennas (RPA's) is the Transmission Line Model (LT) [27]. Equations (1)–(4) describe the LT method and allow one to calculate the parameters of a conventional patch antenna design, i.e., resonant frequency of the dominant mode $f_{010}$, length extension of the patch $\Delta L$, effective dielectric constant $\epsilon_{r_{eff}}$, patch width, and patch length. This method requires previous information about the dielectric constant $\epsilon_r$, the thickness of the substrate $h$, and the antenna frequency $f_r$.

$$W = \frac{1}{2 f_r \sqrt{\mu_0 \epsilon_0}} \sqrt{\frac{2}{\epsilon_r + 1}} \tag{1}$$

$$\epsilon_{reff} = \frac{\epsilon_r + 1}{2} + \frac{\epsilon_r - 1}{2} \left( 1 + 12 \frac{h}{W} \right)^{-1/2} \tag{2}$$

$$\Delta L = 0.412 h \frac{(\epsilon_{reff} + 0.3)\left(\frac{W}{h} + 0.264\right)}{(\epsilon_{reff} - 0.258)\left(\frac{W}{h} + 0.8\right)} \tag{3}$$

$$L = \frac{1}{2 f_r \sqrt{\epsilon_{reff}} \sqrt{\mu_0 \epsilon_0}} - 2\Delta L \tag{4}$$

In the equations above, $W$, $\epsilon_{r_{eff}}$, $\Delta L$, $L$, and $1/\sqrt{\mu_0 \epsilon_0}$ represents the patch width, the effective dielectric constant (taking into account the surrounding space), the patch length extension due to fringing effect, the physical length of the patch, and the speed of light, respectively. We present the geometry of the proposed antenna where the conventional (reference) and the slot-loaded antennas are displayed on Figure 1a and Figure 1b, respectively. In order to set the resonant frequency within the ISM radio band, which allows the antenna to handle many medical and industrial applications, we defined 2.41 GHz as the operation frequency of the dominant mode ($f_{010}$). The dimensions of the antenna were met by calculating Equations (1)–(4). We employed electrical parameters of a fiber-glass laminate substrate FR4 to design the antennas. This material has a nominal dielectric constant $\epsilon_r$ of 4.3, a height $h$ of 1.6 mm, and top and bottom copper cladding thicknesses $t$ of 0.035 mm. Additionally, the feeding line was a microstrip transmission line with a characteristic impedance of 50 Ω. Therefore, we implemented a Quarter-Wavelength impedance transformer (QWIT) for impedance matching between the antenna and the transmission line, and the input impedance of the antenna model was $Zin_0 = (47.80 - j0.2)\,\Omega$ [28]. We modeled the conventional antenna using Computer Simulation Technology (CST), which is a high-performance software for electromagnetic analysis. The fact that the dielectric constant varies as the frequency increases, which occurs in low-cost substrate materials such as FR4, makes us expect the real dielectric value to be different from the nominal value. Hence, it became necessary to establish the actual dielectric constant that would allow the simulation to match the measurements. Therefore, we constructed a first prototype of a conventional patch antenna and then performed a parametric simulation of $\varepsilon_r$. It allowed us to achieve the real dielectric constant $\epsilon_{r_{fixed}}$ of 4.08, i.e., 0.22 (5%) less than the substrate specifications. Based on those results, the dielectric constant $\epsilon_{r_{fixed}}$ was used in the subsequent simulations. In Figure 2, we present a comparison between the measurements and the simulation of the reflection coefficient of the reference antenna. This figure includes the electrical response of the antenna with the nominal and fixed dielectric constant both before and after employing the impedance matching technique via QWT.

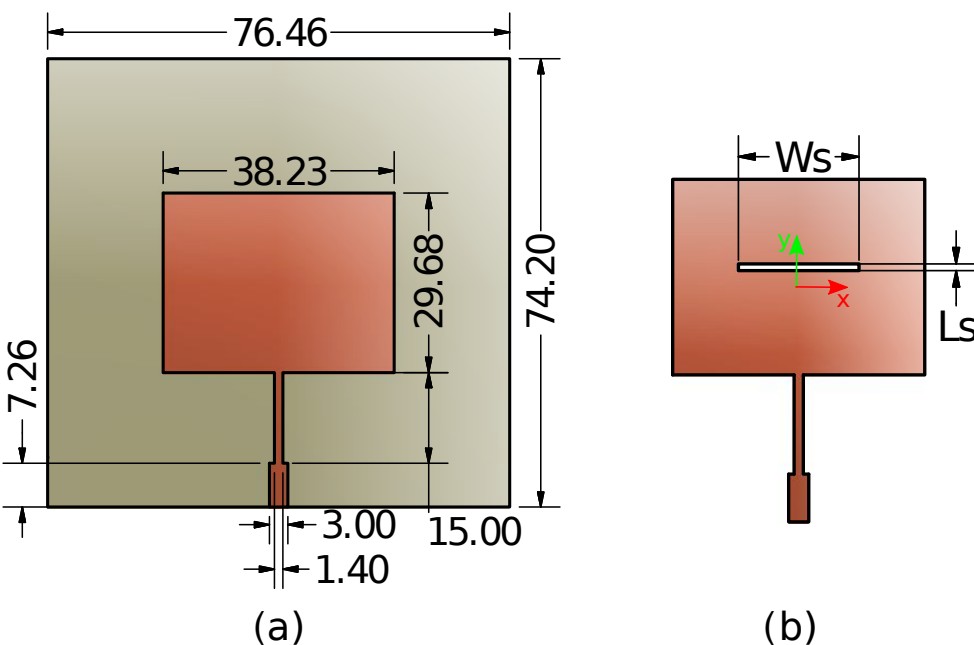

**Figure 1.** Patch antenna design: (**a**) reference patch antenna before slot modification, and (**b**) modified radiating patch with a rectangular slot.

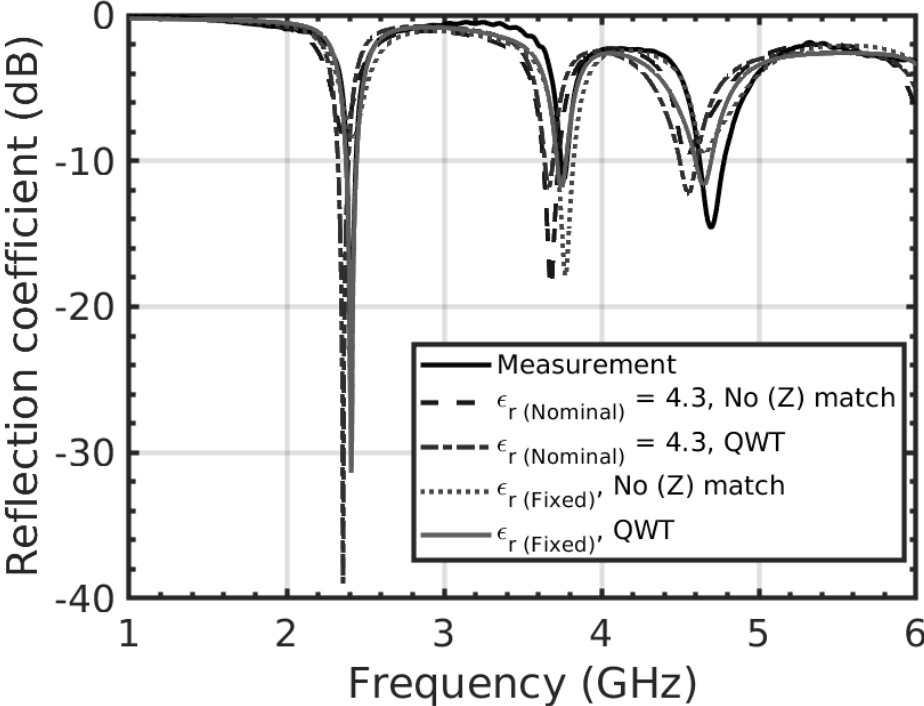

**Figure 2.** Reflection coefficient of the patch antenna before and after the matching impedance and the $S11$ difference between real ($\epsilon_{r_{fixed}}$) and nominal ($\epsilon_r$) dielectric constant.

*2.2. Parametric Analysis*

This section presents a systematic analysis of impacts of the slot on antenna behavior. Hence, we aim to clarify the influence of the slot on several parameters of the antenna, such as the resonant frequency, the reflection coefficient, and the input impedance. Since the path of the surface current for dominant mode must follow only the $y$-axis, the slot represents a discontinuity that will disrupt the current flow in that direction. Thus, the surface current and the electric fields are studied to delve into changes in the antenna's performance.

First, we defined the slot geometry for the initial numerical analysis. As a result, the slot was placed near the origin of the $x$-$y$ coordinates, exactly at $x = 0$ mm and $y = 1.5$ mm. Additionally, the slot length ($L_s$) was defined as a constant 1 mm. Then, a parameter sweep simulation was performed by changing the width of the slot ($W_s$) from 2.2 mm to 18.2 mm, resulting in nine steps. Results suggest a clear trend: both the resonant frequency and the reflection coefficient magnitude are displaced due to an increase in slot width, as shown in Figure 3. A second parametric simulation was performed with regard to the changes in the $y$-position ($y_s$) of the slot and the slot width. Here, the slot length and $x$-position remained at 1 mm and 0 mm, respectively. We prevented the slot from reaching the radiating edges of the dominant mode $TM_{010}^x$; therefore, values from $-13.5$ mm to 13.5 mm in steps of 3 mm (ten steps, and 1.3 mm away from each radiating edge) were given to $y_s$. For every slot position, we established the slot width $W_s$ at four different values: 10.2 mm, 14.2 mm, 16.2 mm, and 18.2 mm. Overall, there were forty iterations to be performed.

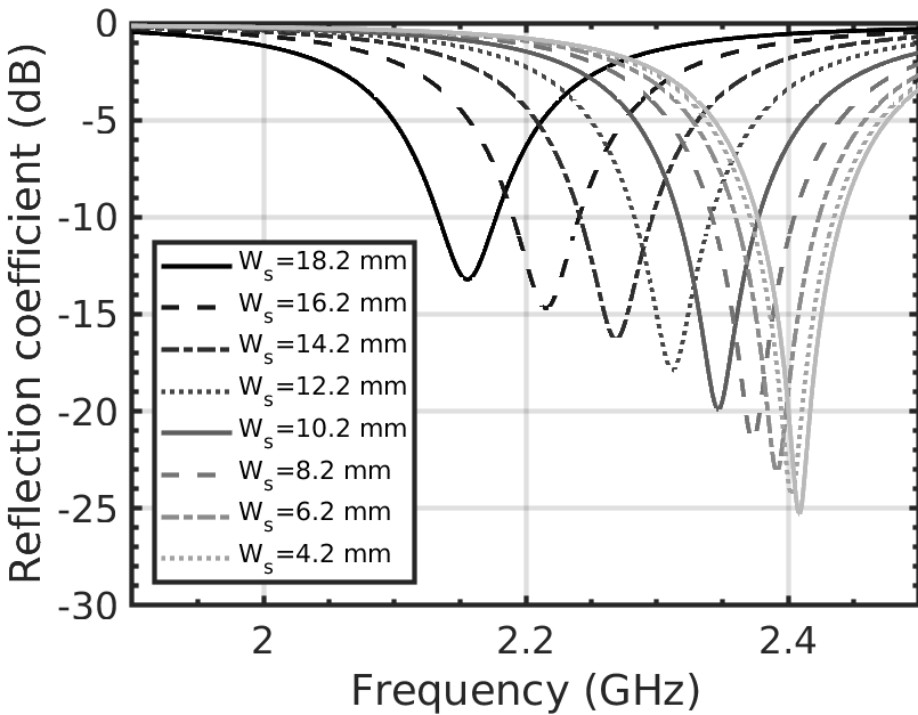

**Figure 3.** Changes in the reflection coefficient ($S11$) due to the variation in slot width.

In the third simulation stage, we analyzed changes in input the impedance ($Z_{in}$), the surface current, and the electric field due to the slot. These analyses were only concerned with variations in the y-slot position, i.e., $y_s = -10.5$ mm, $-7.5$ mm, $-1.5$ mm, 4.5 mm, and 10.5 mm, whereas the slot width remained fixed at $W_s = 18.23$ mm. Since the antennas were going to be experimentally connected to a spectrum analyzer through an SMA to N adapter (which increases the transmission line (TL) length), it was necessary to add 7 mm to the LT of the antenna model (only for simulation concerns), bearing in mind that a TL behaves like an impedance transformer. Regarding antennas parameters, we calculated the surface current at non-radiating antenna edges ($x = \pm 19.115$ mm) and the center of the patch ($x = 0$ mm) for the conventional antenna. Additionally, we calculated the surface current at non-radiating edges ($x = \pm 19.115$ mm), the slot edges ($x = \pm 9.115$ mm), and the center of the patch (through the slot $x = 0$ mm) for the slot-loaded antenna.

Regarding the electric field, we knew that $TM_{010}^x$ mode leads to a field configuration only with z-component within the patch antenna cavity. Other field-components are null according the cavity model. We have performed simulation to calculate the electric field within the antenna cavity. In order to do so, the electric field monitor was defined to

measure electric field along $x = 0$ mm, and considering three slot positions ($y_s = -7.5$, $y_s = -1.5$, and $y_s = 10.5$ mm).

### 2.3. Fabrication and Experimental Setup

In order to experimentally evaluate the influence of the slot on the behavior of the antenna, five antenna prototypes were constructed. We chose antenna models with the slot $y$-positions ($y_s$) at $-10.5$ mm, $-7.5$ mm, $-1.5$ mm, $4.5$ mm, and $10.5$ mm. So we tried to cover all the slot positions along the patch length. Additionally, the slot width $W_s$ remained at $18.2$ mm. To make a comparison, we built a reference antenna (without the slot) as well. The antenna models were saved as G-code commands required for Computer Numerical Control (CNC). In the CNC process, the radiating patches and transmission lines were cut using an end mill of $0.2$ mm in diameter, and the perimeter was cut using an end drill of 1 mm in diameter. Additionally, for milling, we adjusted the penetration depth to approximately $0.035$ mm. Therefore, the contours of the radiating patch and the transmission line guaranteed accurate properties such as straight side walls and small penetration depth into the substrate surface. We welded an SMA connector of $50\,\Omega$ of impedance onto each antenna prototype. Subsequently, we connected the antennas to a Vector Network Analyzer (VNA) through an SMA-to-N adapter to take measurements of the resonant frequency, the input impedance, and the magnitude of the reflection coefficient on each antenna. In Figure 4 and Figure 5, respectively, we present the manufactured antennas and the experimental set up that we adopted. The following sections compare the results of the simulations and the measurements.

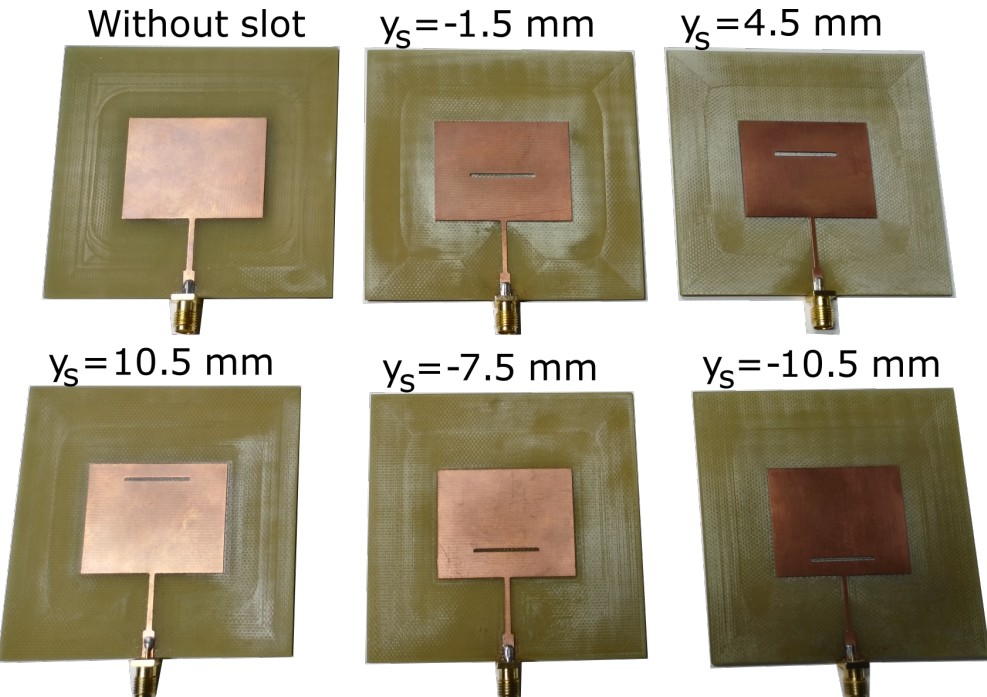

**Figure 4.** Antenna prototypes with 18.2-mm wide slots.

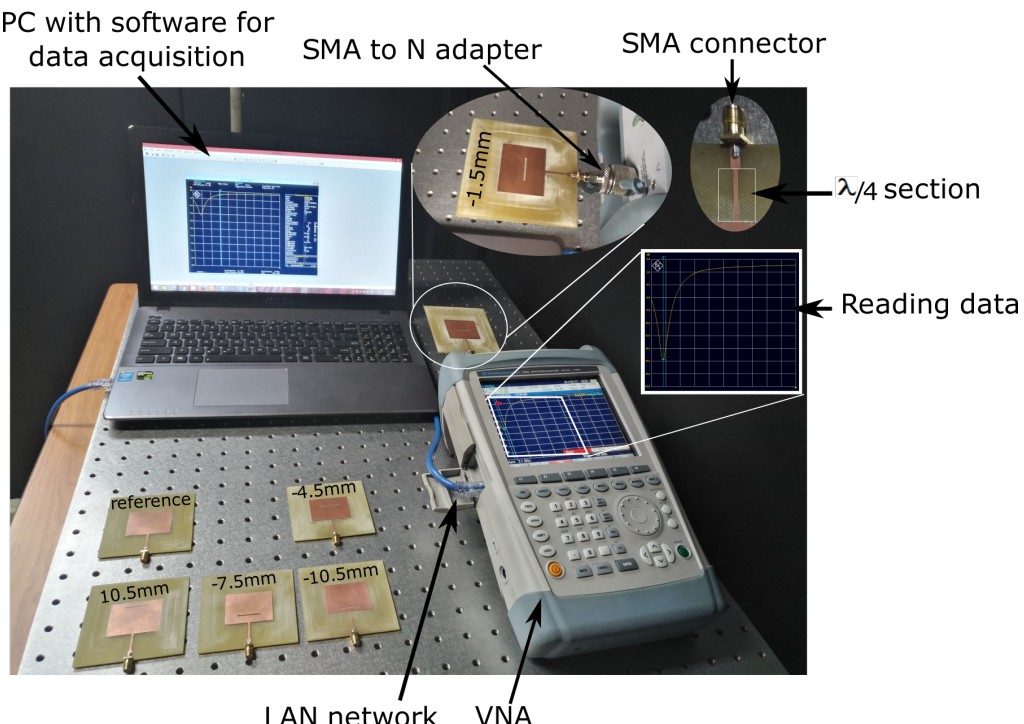

**Figure 5.** Experimental setup.

## 3. Results

In this section, we describe the results of the simulation and the measurements obtained. Additionally, here we explain the meaning of the results and try to find a physical explanation for them.

### 3.1. Simulation Results

3.1.1. Influence of the Slot on Resonant Frequency, Reflection Coefficient, and Input Impedance

There exists an inverse relationship between the resonant frequency of the antennas and the slot width, as shown in Figure 6, where the resonant frequency shifts towards lower values as the slot becomes wider. Furthermore, the slot position along the $y$-axis causes noticeable changes in the resonant frequency ($f_{010s}$), as it is shown in Figure 6. The resonant frequency suffers minimal changes at the upper and lower limits of the slot $y$-position, $y_s = -13.5$ mm and $y_s = 13.5$ mm, respectively. Preceding means that slot being close to either radiating edges ($TMx_{010}$) had imperceptible influence on the resonant frequency, so it stands around the initial value ($f_{010} = 2.41$ GHz). Contrarily, as the slot moves towards the center of the patch, the resonant frequency gradually decreases, and the minimum value of it (i.e., maximum frequency shift from $f_{010}$) occurs when the slot position is around the middle of the patch (around zero $y$-position). Furthermore, changes in the resonant frequency also depends on the slot width at each slot position. As a result, when the slot width is set to its maximum, the shift in the resonant frequency is maximum as well. In contrast, when the slot is narrow, the resonant frequency shifts minimally.

From previous results, we derived models which describe the behavior of resonant frequency as a function of the slot position and width. The resonant frequency $f_{010s}$ resulted in a superposition of the initial resonant frequency and a cosine function scaled in amplitude and phase by the slot width and slot $y$-position, respectively. The model equation and its parameters can be found in Equation (5) and Table 1, respectively. In addition, we found that all the prediction models exhibited an error beneath 11.5 MHz.

$$f_{010s} = f_{010} - a_{W_s} \cos\left(0.116\, y_s\right) \tag{5}$$

where the quantity (0.116) inside the argument of the cosine function has a magnitude of 1/mm.

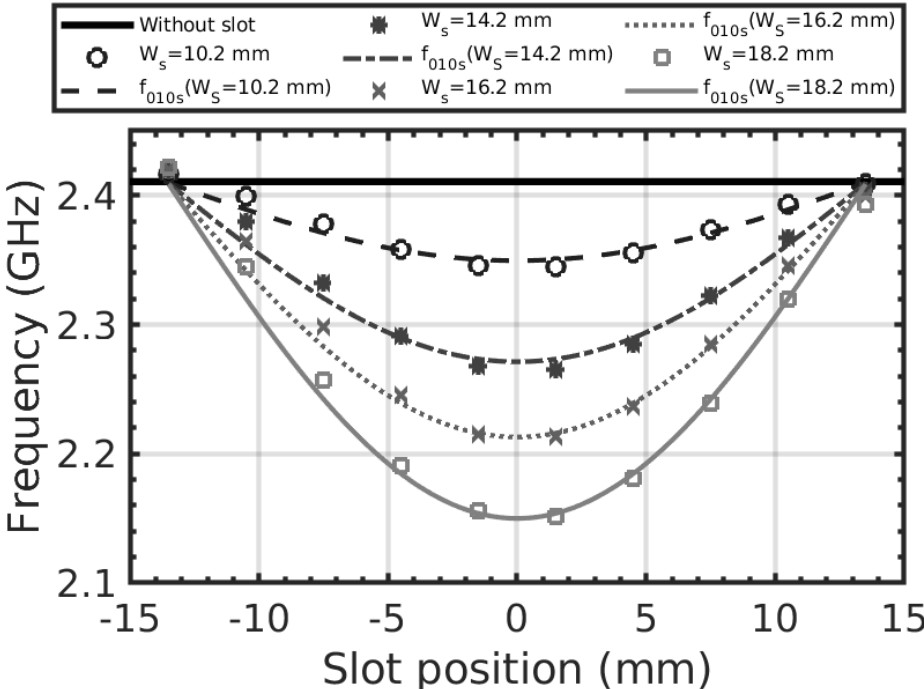

**Figure 6.** Resonant frequency changes due to different slot positions along the length of the patch and curves fitting.

**Table 1.** Coefficients of frequency model (Equation (5)).

| Model | $W_s$ (mm) | $a_{W_s}$ (GHz) | $f_{010}$ (GHz) | RMSE (MHz) | $R^2$ |
|---|---|---|---|---|---|
| $f_{010s}$ | 10.2 | $-0.060$ | 2.41 | 5.38 | 0.96 |
| | 14.2 | $-0.139$ | 2.41 | 8.98 | 0.97 |
| | 16.2 | $-0.197$ | 2.41 | 10.76 | 0.98 |
| | 18.2 | $-0.260$ | 2.41 | 11.49 | 0.99 |

Let us now consider the changes in the magnitude of the reflection coefficient $|\Gamma|$ due to slot position and width. Results in Figure 7 are similar to those mentioned earlier regarding the resonant frequency in the fact that there is a region with a greater influence on the antenna's behavior. Let us define four point/region for slot position: (1) at $-13.5$, (2) from $-10.5$ to $-4.5$, (3) at $-1.5$ mm, and (4) form $-1.5$ to 13.5 mm. Hence, the $|\Gamma|$ reaches its middle value when the slot is at each point (1) and point (3), and it reaches its maximum value when slot is within region (2). Finally, all $|\Gamma|$ values lean over the reference value of the antenna without the slot as $y_s$ increase.

Accordingly, the slot width has a significant effect on $|\Gamma|$ as long as the slot is located within the range from $-13.5$ to 4.5 mm. However, slots located nearly the second radiating edge ($y_s = 13.5$ mm), the slot width produces minor effects, and all of the $|\Gamma|$s converge to the reflection coefficient amplitude of the reference antenna $|\Gamma|_0 = -28.92$. We derived an analytic expression that describes the relationship between the reflection coefficient magnitude and the slot position and width. This prediction model is shown in Equation (6), and its parameters are listed in Table 2.

$$|\Gamma_s| = |\Gamma_0| + c_{W_s} \left[ \sin\left(0.1052 \, y_s + 1.666\right) \right] e^{-0.1052 \, y_s} \tag{6}$$

where the quantities (0.1052 and 1.666) inside the argument of the sine and exponential functions have a magnitude of 1/mm.

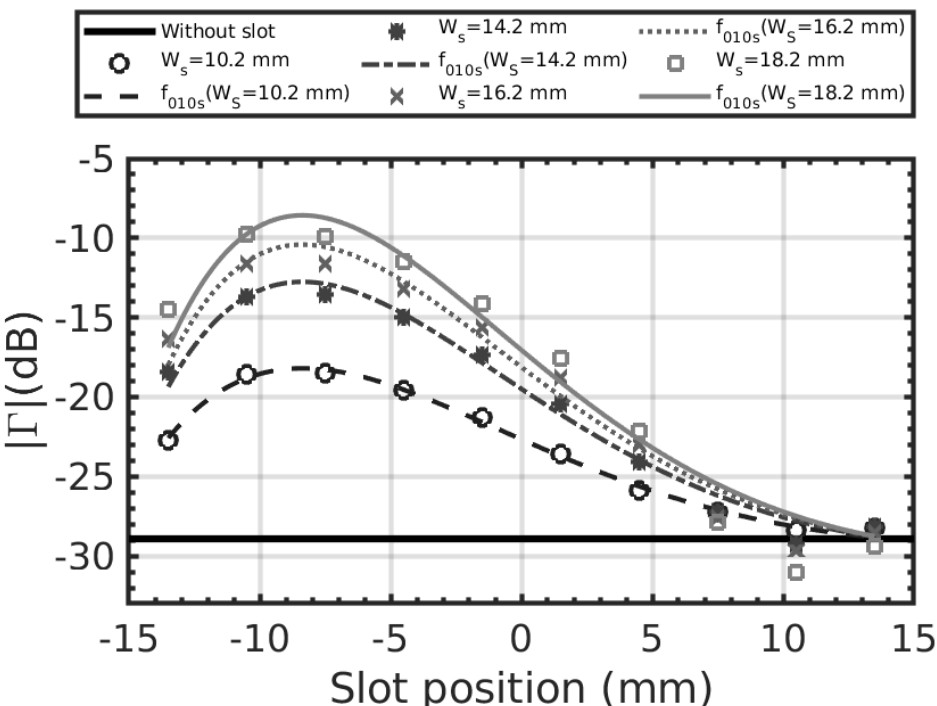

**Figure 7.** Reflection coefficient amplitude changes due to slot position and curves fitting.

**Table 2.** Coefficients of the amplitude reflection model.

| Model | $W_s$ (mm) | $c_{W_s}$ (dB) | $|\Gamma_0|$ (dB) | RMSE (dB) | $R^2$ |
|---|---|---|---|---|---|
| $|\Gamma_s|$ | 10.2 | 6.28 | −28.92 | 0.28 | 0.99 |
| | 14.2 | 9.47 | −28.92 | 0.79 | 0.98 |
| | 16.2 | 10.85 | −28.92 | 1.22 | 0.97 |
| | 18.2 | 11.92 | −28.92 | 1.79 | 0.95 |

Moving on now to consider the influence of the presence of the slot on the input impedance, Figure 8 shows that both the resistance and the reactance of the input impedance were affected by the slot. These parameters were only studied at a slot width of 18.2 mm. From the results, it can be seen that, as the slot approaches the feeding edge, the input impedance leaves the optimum resonance condition $(50 + j0)\,\Omega$. In turn, when the slot position remains in the range from 7.5 to 13.5 mm, the input impedance becomes similar to the desired resonance condition. Additionally, Figures 7 and 8 show that there is a trend between $Z_{in}$ and $|\Gamma|$. When $Z_{in}$ is at the resonance condition, $|\Gamma|$ becomes minimum, which is expected since matched networks produce a minimal reflected wave. Finally, we derived a prediction model of the input resistance from the input impedance data. Such a model is presented in expression (7).

$$Zin_s = 10.44\sin\left(0.1819\,y_s\right) + 37.24 \tag{7}$$

Dashed-lines that follow the dispersion data in Figures 6–8 represent the prediction models of the resonant frequency, amplitude of the reflection coefficient, and input impedance for each slot width, respectively. Additionally, the wider solid lines in those figures represent the parameters of the reference antenna design, i.e., $f_{010}$, $|\Gamma_0|$, and $Zin_0$.

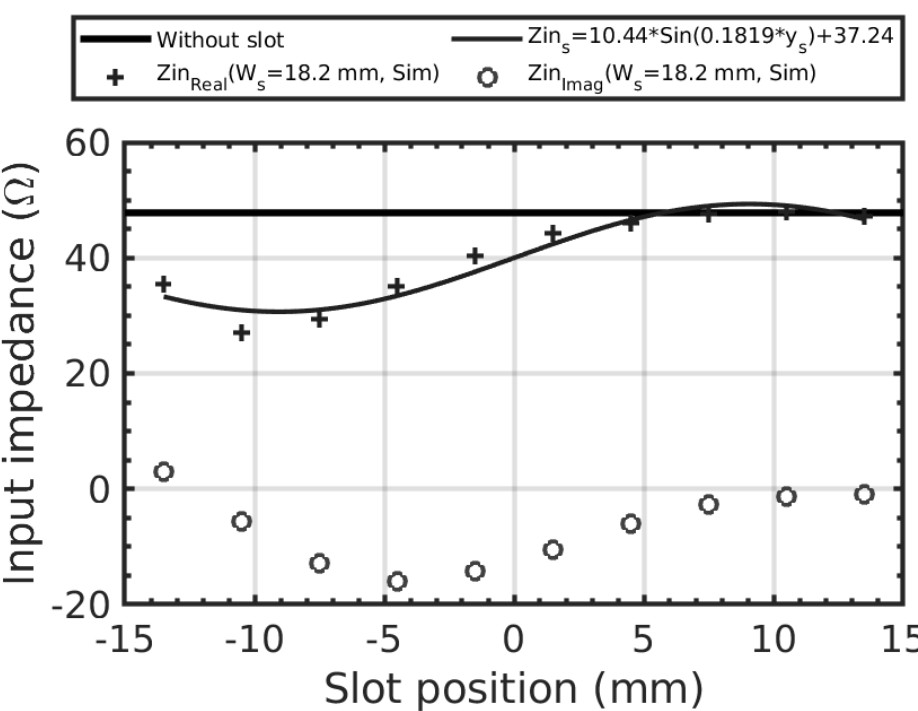

**Figure 8.** Input impedance changes due to the slot variation along the length (*L*) of the patch and curves fitting.

3.1.2. Surface Current and Electric Field

Let us now consider the influence of the slot on the surface current and the electric field distribution of the antenna. Figure 9 shows how the slot changes the current and the electric field. Before the slot was introduced, the surface current and the electric field corresponded exactly to those described for the dominant mode $TM_{010}^{x}$ of the cavity model, as can be seen from Figure 9a. In the case of the dominant mode, the structure has only y-surface current and z-electric field components. However, when the slot is introduced, the surface current tends to surround it, and the electric field has not only the z-component, but also $x$ and $y$ components, as shown in Figure 9b,c. We used Figure 10 to quantitatively describe the surface current on the radiating patch. These analyses were focused on the y-surface current behavior at $x = w/2$, $-w/2$ (both non-radiating edges), $x = 0$ mm (the center of the patch), and at both edges of the slot in the slotted antennas. The surface current presents a high concentration toward the non-radiating edges and the slot's edges, which produces the highest current at the slot's edges. The slot would be a mechanism that allows one to control the resonant frequency of the antenna as well as the reflection coefficient amplitude by changing the distribution of the surface current within the patch. So, when the slot is in the middle of the patch, a considerable amount of the surface current focuses on the slot's edges, and this has the highest influence on the antenna response by shifting the resonant frequency to its lower value. In contrast, while for slot positions nearby to both radiating edges only a few surface current concentrates around the slot, and it has less influence on the resonant frequency variation. In addition, one can infer that at both the upper and lower limit of the slot position, the slot will be out of the patch and there would be a null influence on the antenna reflection coefficient. Thus, the antenna will behave as the reference one, i.e., the one without the slot.

In Figure 11a, we show how the z-electric field component of the dominant mode varies along the length of the patch (*L*). Regarding the electric field, in Figure 11b–d, it can be seen that the z-electric field amplitude changes drastically across the slot. Since the added slot deviates the current path, the electric field would vanish due to the slot presence.

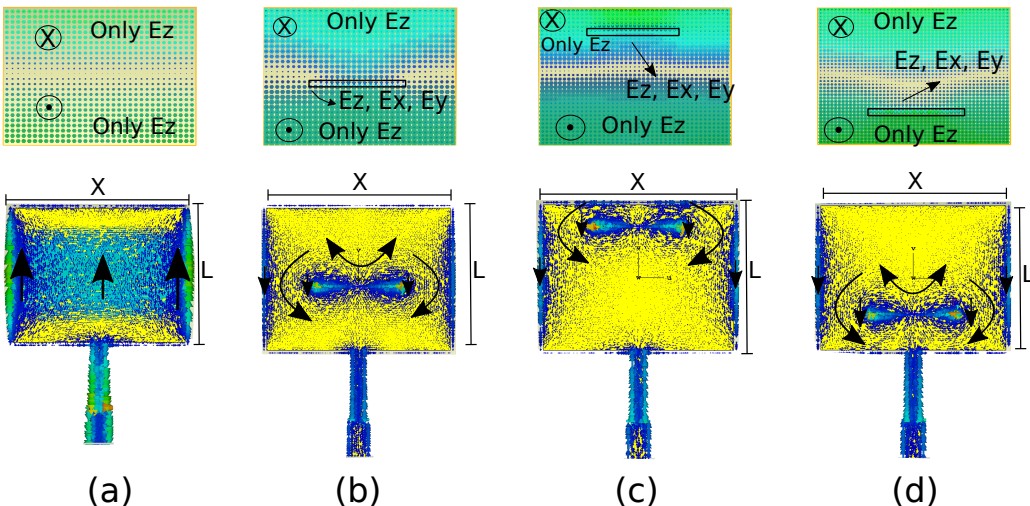

**Figure 9.** Qualitative description of the surface current and the electric field distributions of the antennas. (**a**) the reference, and antennas with the slot in different positions: (**b**) $y_s = -1.5$ mm, (**c**) $y_s = 10.5$ mm, and (**d**) $y_s = -7.5$ mm.

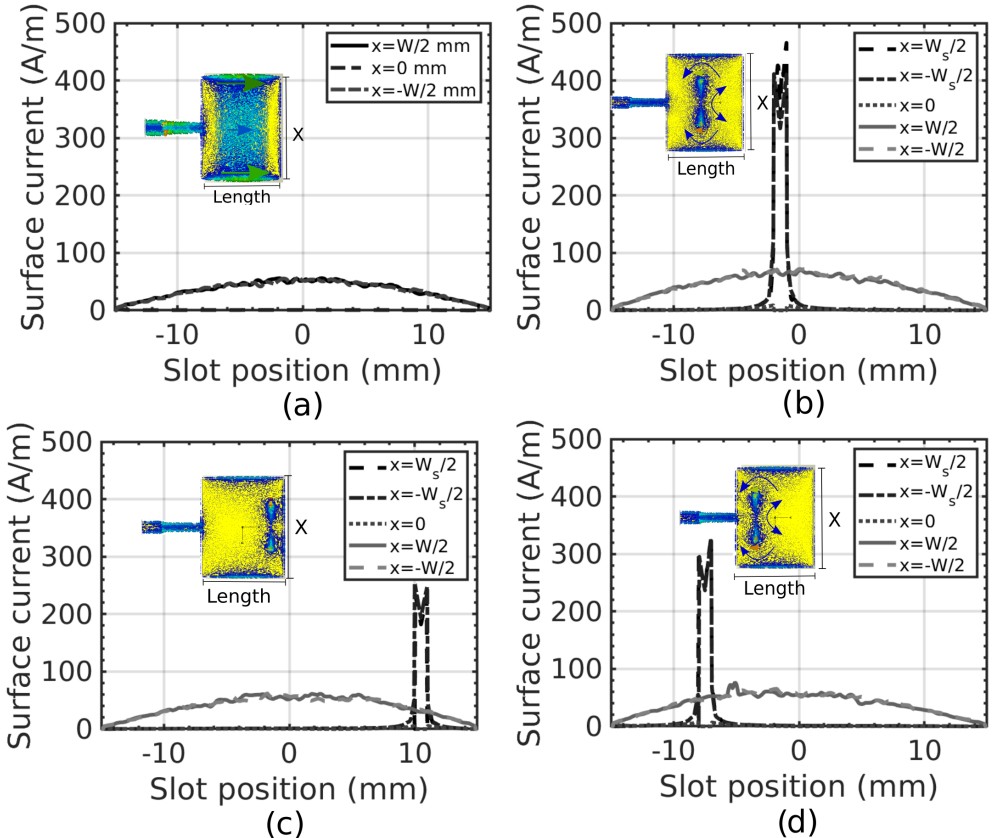

**Figure 10.** Surface current on the radiating patches of (**a**) the reference antenna and antennas with the slot in different positions: (**b**) $y_s = -1.5$ mm, (**c**) $y_s = 10.5$ mm, and (**d**) $y_s = -7.5$ mm.

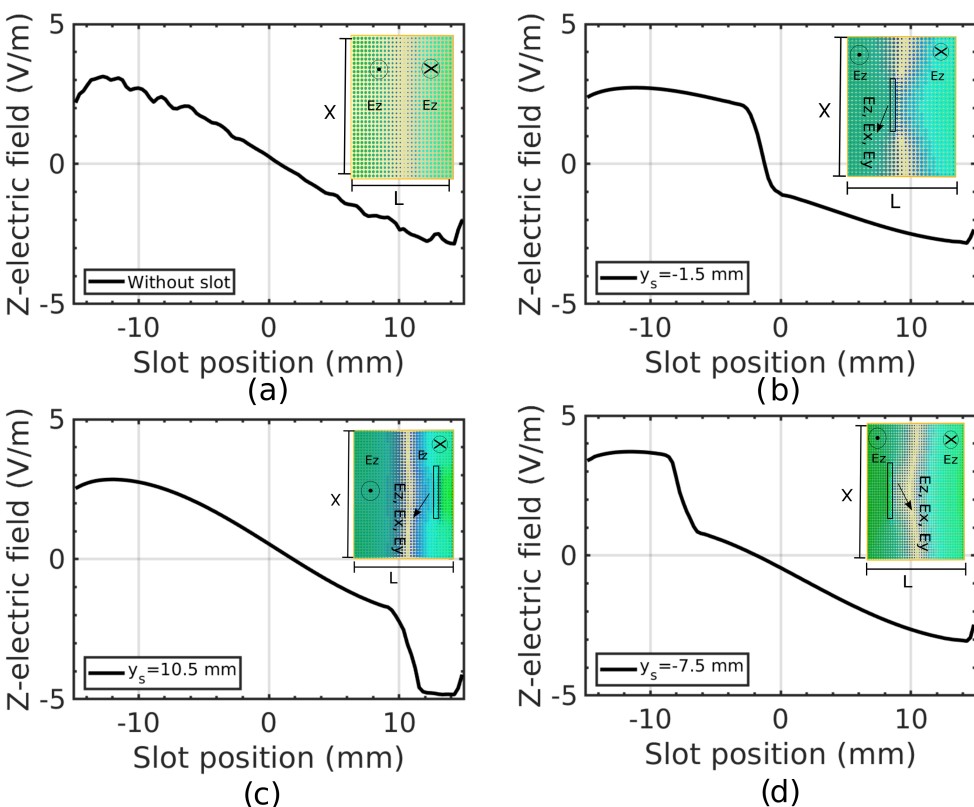

**Figure 11.** Electric field distribution on the radiating patches of (**a**) the reference antenna and antennas with the slot in different positions: (**b**) $y_s = -1.5$ mm, (**c**) $y_s = 10.5$ mm, and (**d**) $y_s = -7.5$ mm.

*3.2. Comparison between Numerical and Experimental Results*

In this section, we present a comparison between numerical results and measurements that validates the simulation analyses and the relationship between slot geometry (position and width) and electrical antenna response described earlier. In Figure 12, we show both the experimental and numerical results of the resonant frequency as a function of the slot position on the patch ($y_s$). It is apparent that the numerical and experimental results coincide in each slot position, as well as regarding the antenna without any slot. In fact, only a minority of the measurements significantly differed from their simulation counterparts, resulting in a deviation of 0.6 % for $y_s = -1.5$ mm and 1.4 % for $y_s = 4.5$ mm. Turning now to the amplitude of the reflection coefficient, we present a comparison between simulations and measurements as shown in Figure 13. There is a good agreement between simulations and measurements within almost all the range of slot positions ($y_s$). The most significant discrepancy was detected at slot position $y = -1.5$ mm and $y_s = 4.5$ mm, which produced an error of 5.4 % and 5.9 %, respectively. Finally, in Figure 14, we compare simulations and measurements of changes in input impedance due to the slot. The real part of the input impedance shows an agreement between the simulations and the measurements, with just a little error for the antenna with the slot at $-7.5$ mm and at $-1.5$ mm. Additionally, the imaginary part of the input impedance measurements match the simulations in all the range of ($y_s$).

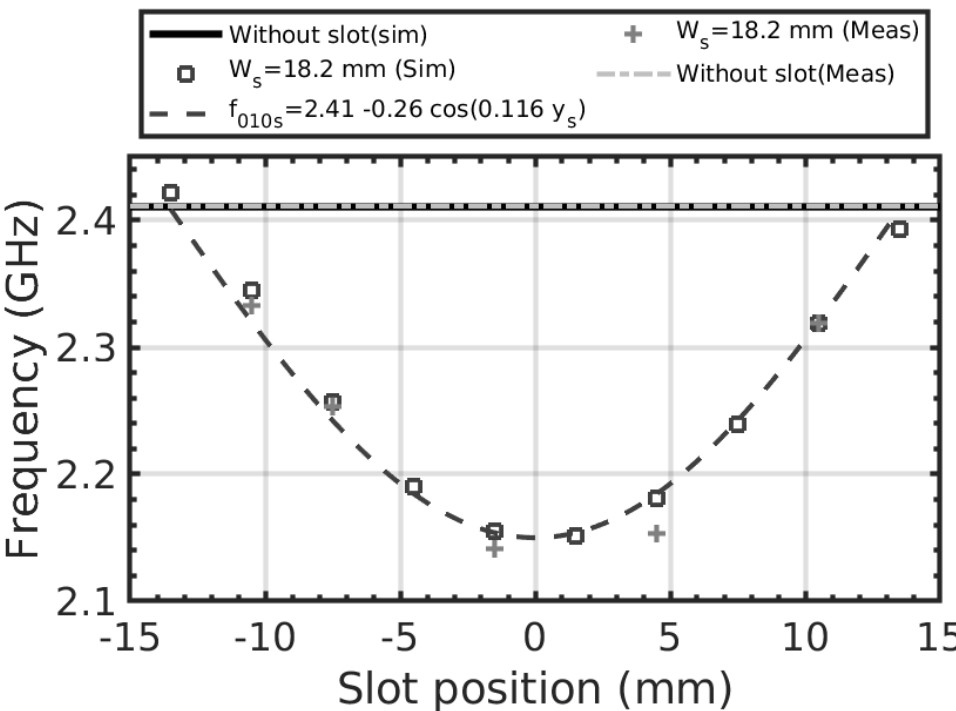

**Figure 12.** Simulation and measurements of resonant frequency changes due to slot position on the patch.

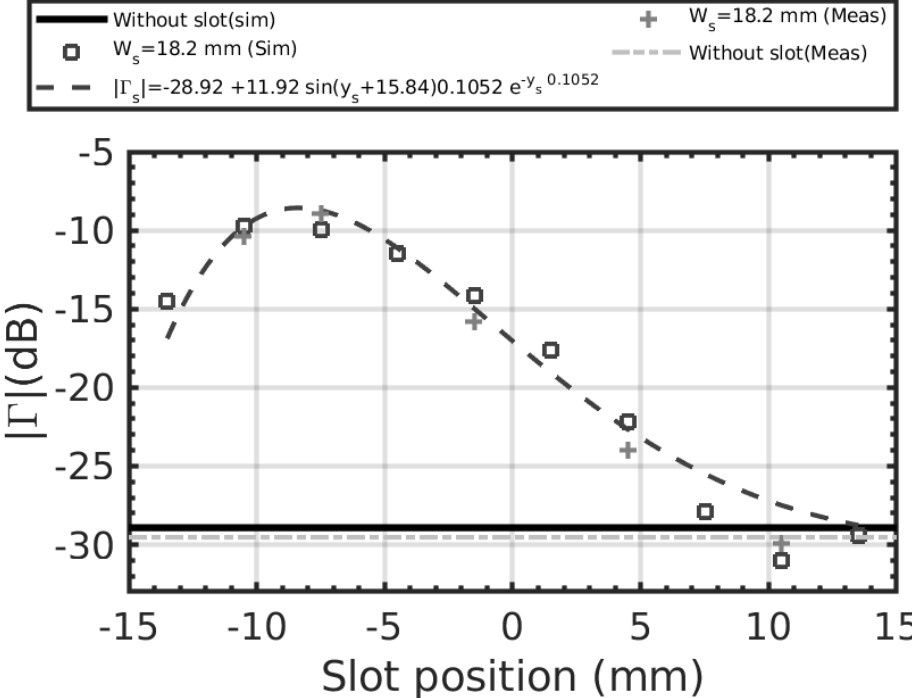

**Figure 13.** Simulation and measurements of S11 amplitude changes due to slot position on the patch.

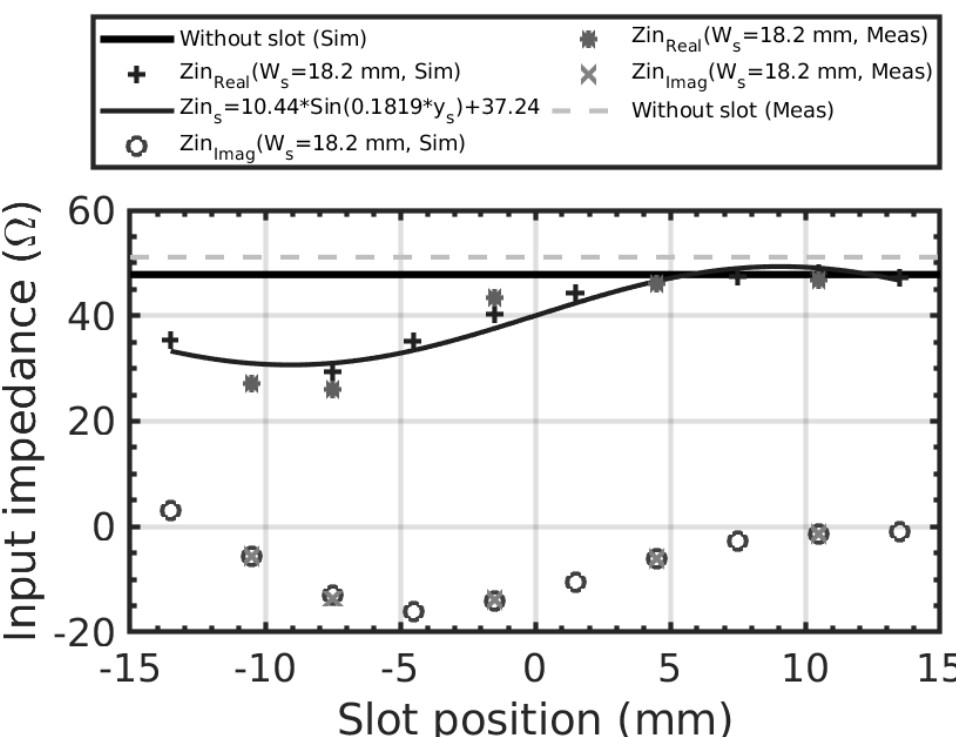

**Figure 14.** Simulation and measurements of the input impedance changes due to slot position on the patch.

## 4. Discussion

Prior studies have reported the importance of slot modifications for antenna design; however, they have not provided analytic expressions that describe the impact of such modifications on the reflection coefficient of the antenna. In fact, the most relevant findings of our work are the analytic models presented in Equations (5) and (6). Those models describe how the dominant resonant frequency $f010_s$ and the amplitude of the reflection coefficient $|\Gamma_s|$ change as a function of a parameterization of the slot position and width. Based on those results, we could claim that there are two mechanisms that influence variations in the resonant frequency and the amplitude of the reflection coefficient: (1) the increase in the electrical length and (2) the fluctuation in the input impedance; both due to the slot. Increasing the electrical length makes the resonant frequency shift toward lower values. With all the slot widths, the resonant frequency dropped suggesting an increase in the effective electrical length of the antenna patch.

Remarkably, when the slot was near both radiating edges, it had very little influence on the resonant frequency. In contrast, as the slot approaches the center of the radiating patch, the change in resonant frequency reached its maximum. A reasonable explanation for this behavior might be that the current distribution follows the sine function $J = \sin\left(\frac{\pi x}{L}\right)$, which has its maximum in the middle of the patch ($L/2$). Therefore, when the slot is located around the center of the patch, it represents a significant obstacle for the current path. In contrast, the resonant frequency would have similar value to the antenna without the slot (a rectangular patch) when the slot is located at either the upper or lower radiating edges. Hence, the resonant frequency of the antennas with the slots in their limit positions would be expected to be the same as one of the antenna without the slot.

On the other hand, the variation in the input impedance due to the slot position has a substantial connection with the amplitude of the reflection coefficient of the antenna. Negative slot *y*-positions influence larger variations in the input impedance (real and imaginary part), which cause the reflection coefficient to increase. On the contrary, as the

slot *y*-position becomes positive, i.e., close to the second radiating edge, the $|\Gamma_s|$ tries to return to its reference value (antenna without the slot).

Regarding the electric field and the surface current, the slot produced abrupt changes in the path of the surface current, which changes the electric field configuration. So, around the slot there exist not only x and y, but also a z-component of the electric field. We deem that those results are a consequence of the changes in the surface current path due to the slot and the high current concentration around it.

Based on the cavity model applied to patch antennas, it is known that there exists only a z-component of the electric field within the cavity, which has a distribution described by cos(pi*x/L). Therefore, the strongest z-component of the electric field would be around the radiating edges ($y_s = -L/2$ and $y_S = L/2$) of the patch, and a slot placed in such a position would have a high influence on the electric field. On the contrary, a slot placed around the middle of the patch ($y_s = 0$) would have minimal influence on the electric field. Since there is a relationship between the electric field and the resonant frequency of the patch antenna, the aforementioned trade-off between the electric field and the slot position could be extended to the relation between the slot position and the resonant frequency. In fact, our results showed that the shift of resonant frequency is proportional to a cosine function, as is shown in Equation (5). Our results can be helpful to the improved understanding of modified antennas design for improving applications such as gain enhancement [29], phase correction [30], and beamsteering [31].

As indicated previously, the model in Equation (5) represented shifts in the frequency as the initial resonant frequency $f_{010}$ plus a harmonic function perturbation. We found a parallel situation regarding the amplitude of reflection coefficient: its changes obeyed the $|\Gamma_0|$ value of the original antennas plus a harmonic and an exponential function perturbation. Moreover, Equations (5) and (6) establish that those models can be used to find specific electrical parameters of a patch antenna without using any numerical software that implements a more complex solution method (such as finite elements) and requires expensive hardware resources and long execution times.

Although Equation (5) does not involve physical parameters such as capacitance and inductance of the slot, it does accurately describe the experimental results taking into account the geometric characteristics of the slot, and it is an important parameter in antenna design. Furthermore, the slot describes the modifications in the Ez component of the electric field within the antenna (Figure 11), which in turn control the resonance frequency (and other parameters, Figures 12–14). On the other hand, the proposed model can provide relationships beyond the known analytical models and take into account new parameters like the slot width and length that are not addressed in conventional models. Moreover, the model corresponds to the experimental results with almost negligible error, and it reaches results similar to those found with the models of transmission lines and resonant cavities, i.e., in the case when the dimensions of the slot approach zero.

In addition, our results addressed some interesting questions regarding slot antenna analyses asked by (Joler, 2015), which confirm the importance of understanding the relationship between the frequency and antenna parameters such as the length, width, or position of the slot.

## 5. Conclusions

This paper discussed the influence of slot parameter variations on the resonant frequency, reflection coefficient amplitude, and input impedance of a patch antenna. Analytical models of the influence of the slot on the rectangular microstrip patch antenna were developed, which was the main goal of this study. Those models allowed us to calculate the resonant frequency of the dominant ($TM_{010}$) mode and the amplitude of the reflection coefficient of a slotted antenna with several slot positions and widths. The proposed models were verified using measurements which showed good agreement. Additionally, the models enabled a reliable and quick computation of the antenna's parameters using a simple calculation.

The resonant frequency, reflection coefficient amplitude, surface current, and electric field distribution of the slotted antenna depend on the slot position. When the slot is at its upper limit, the resonant frequency and the amplitude of the reflection coefficient tend to be the same as those of the antenna without the slot. In addition, the resonant frequency of the slotted antenna and the reference antenna are alike even when the slot is at its lower limit (nearby the feeding edge). Regarding the reflection coefficient, it becomes maximum when the slot is near the feeding edge, which results in a major impedance mismatch. In every position, the slot presents an obstruction for the surface current path, which influences increments in the electrical length, changes in the resonant frequency, and the occurrence of new electric field components.

## 6. Future Work

The questions raised by this study are: What is the actual relationship between the antenna modification (by adding a slot) and antenna parameters such as the electric field and the surface current? How do those modifications influence the resonant frequency to shift? Characteristic mode theory may be used to address these issues. In addition, further research might explore 2D models for calculating antenna parameters taking into account the relationship with the physical parameters of the system such as the capacitance and the inductance of the slot, including scalability to different substrates and different $\varepsilon_r$, and extending our analysis to multiple band antennas for sensing spectrum applications in cognitive radio systems.

**Author Contributions:** Conceptualization, O.O.-M. and F.L.-G.; methodology, O.O.-M. and F.L.-G.; software, O.O.-M.; validation, O.O.-M.; formal analysis, O.O.-M. and F.L.-G.; investigation, O.O.-M. and F.L.-G.; resources, F.L.-G.; data curation, O.O.-M.; writing—original draft preparation, O.O.-M. and F.L.-G.; writing—review and editing, O.O.-M. and F.L.-G.; visualization, O.O.-M. and F.L.-G.; supervision, O.O.-M. and F.L.-G.; project administration, O.O.-M. and F.L.-G.; funding acquisition, F.L.-G. All authors have read and agreed to the published version of the manuscript.

**Funding:** This research received no external funding.

**Acknowledgments:** The authors would like to thank Institución Universitaria Pascual Bravo (Medellín, Colombia) and Instituto Tecnolóogico Metropolitano (Medellín, Colombia) for their financial support and support with equipment for taking measurements. Additionally, the authors would like to thank engineer Gabriel Palacio for prototyping and constructing the antennas.

**Conflicts of Interest:** The authors declare no conflict of interest.

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
