# Peer review of "A Simple Model to Compute the Characteristic Parameters of a Slotted Rectangular Microstrip Patch Antenna"

_electronics, doi:10.3390/electronics11010129_

Round 1

Reviewer 1 Report

This paper presents an analytical model of a slot-loaded patch and compares it to experimental results. The paper is interesting but the following comments need to be addressed, as the proposed model cannot be described as an analytical model when it’s primarily based on numerical simulations.

-The proposed equation (5), the main focus point of the work, is unclear. It appears to me that a number of CST simulations were used and fitted using an analytical curve fitting technique. The regulating sine/exponential formula is not derived based on physical parameters, e.g. the capacitance and inductance of the slot.

-Following on the entirely curve-fitted solution in (5), the expression for the reflection coefficient (6) is even more complex and further away from an “analytical solution” which is derived from the antenna’s equivalent circuit/transmission line model.

-The proposed (fitted) equations (5) and (6) were only investigated for a fixed relative permittivity. As (5,6) don’t include a term for E_r, their scalability to different substrates and different E_r needs to be considered.

-Following on the pervious question, what is the effect of the tan-delta on the computed reflection coefficient? It is expected that at least the magnitude will change and the proposed formulas do not reflect that.

-The abstract could benefit from more quantified results, for example on the discrepancy between the analytical and measured/3D EM calculation.

-The section titles are not indicative of the paper’s content and aren’t very well suited to this type of research. In particular, “Materials & Methods” isn’t relevant to this work; it is advised to follow the naming convention of similar articles in this field.

-A discussion of the slot’s impact on the far-field will be very helpful to make the study more complete, given that CST-simulated E-field/surface current plots are already presented.

-The paper needs to be carefully revised for typographical and grammatical errors, with lots of incomplete sentences. e.g. line 18 “allowed to proposed primarily”. Also, the Figures format and size needs to be revised to enhance the paper’s claity 

Author Response

Dear Reviewer,

We are grateful to you for your valuable comments and suggestions. We have analyzed the comments and made modifications accordingly. The authors hope the revised manuscript meets with approval. Changes have been highlighted (Green color) in the revised manuscript, while our responses were highlighted with blue color. 

Reviewer 2 Report

This manuscript presents A Simple Model to Compute the Characteristic Parameters of a Slotted Rectangular Microstrip Patch Antenna. The work is interesting, but some information’s are missing from the manuscript that is very important to the readers. Some queries and suggestions are given in the following points for the betterment of the manuscript. Please include the findings or novelty of the proposed design in the abstract section to show the reader's particular contribution

  1. Page 1. Introduction 1st line: please insert “of” to make “slot type of
  2. Page 1. 2nd column 7th last line: please replace of with “on” to make “A study on the impact of..”
  3. Page 2. 2nd column 13th line: please rephrase the sentence “Among the properties of the fibre-glass substrate= material (FR4) that we considered in the design process there was a nominal dielectric constant of 3, a height of 4: h mm, and top and bottom copper cladding thicknesses 1:6 t of mm 0:035”
  4. Page 4. 2nd column, 22th line: please insert “equation” to make “parameters can be found in equation 5.
  5. Page 4. 1st column, result and discussion: please put a full stop at the end of the paragraph.
  6. Page 5. 2nd column, 16th last line: please replace figure 9 with “fig. 9”
  7. Page 8. 1st column 8th line: please correct the spelling of even.
  8. Abbreviations of figure is mentioned differently in a different part of the document. Please use one preferably as fig. 1 .

Author Response

(The authors gave the same response as above.)

Reviewer 3 Report

This paper developed an analytical model of slot-loaded rectangular microstrip patch antennas based on the simulation results by varying slot parameters. But this work is lack of novelty. And if the model can be used to multi-slot and different slot geometry? How applicable is the model? 

Author Response

(The authors gave the same response as above.)

Reviewer 4 Report

This work provides an insightful analysis regarding the effects of slot rectangular. While using slot resonators is not a new concept in the antenna community, authors have done a great job developing an analytical model to explain and control the antenna outputs. There some aspects that need to be fixed, though as below:

The introduction is a bit monotonous and almost in all cases, author-prominent paraphrases (indirect voice) have been used. Authors should use a range of paraphrasing techniques in writing the introduction to avoid tedious text. In science, external voice (information-prominent paraphrase) is much more preferred. Please improve the writing style of the introduction based on the matters discussed above

The template is not Electronics journal template and needs to be changed.

The works cited in the Intro are not quite new (the newest is 2017). There have been many interesting applications of slots published in recent years (2020 and 2021). Here are some examples that can be added to the intro.

Slots also have been used in flexible antennas to achieve monopole-like radiation patterns and patterns reconfigurability. Achieving monopole-like radiation patterns using slots: A conformal ultrawideband antenna with monopole-like radiation patterns, 2020 by Mohamadzade.

Achieving pattens reconfigurability using slots: A Conformal, Dynamic Pattern-Reconfigurable Antenna Using Conductive Textile-Polymer Composite, 2021. Additionally, pair rectangular slots were recently proposed for extending Axial ratio in patch antennas as explained in: Two-Pair Slots Inserted CP Patch Antenna for Wide Axial Ratio Beamwidth.

Section III. 2 (Electric field distribution): Surface current explanations need an improvement and more elaboration. What is the ideal current distribution in your case?

Please explain the mechanism you captured the electric field and also include other applications of the electric field distribution analysis. Electric Near-field analysis provides an excellent insight into antennas propagation and has wide applications in gain enhancement as explained in (Directivity improvement of a Fabry-Perot cavity antenna by enhancing near field characteristic), phase correction (All-metal wideband metasurface for the near-field transformation of medium-to-high gain electromagnetic sources) and beamsteering (Beam-scanning antenna based on near-electric field phase transformation and refraction of electromagnetic wave through dielectric structures). Adding these applications would add reliability to the electric field analysis used in the paper.

Please explain how you come up with the step size and initial starting point in your parametric sweep?

Author Response

(The authors gave the same response as above.)

Round 2

Reviewer 3 Report

Characteristic mode theory may be used to analyze the problem in this manuscript.

Author Response

Dear reviewer,

We are grateful to you for your valuable comments and suggestions. We have analyzed the comments and made modifications accordingly. The authors hope the revised manuscript meets with approval. Changes have been highlighted (Green color) in the revised manuscript, while the responses were highlighted with (Blue color). 

Reviewer 4 Report

The paper has been improved and all concerns have been addressed.

Author Response

(The authors gave the same response as above.)
